# Modulation of the Aryl Hydrocarbon Receptor Signaling Pathway Impacts on Junín Virus Replication

**DOI:** 10.3390/v15020369

**Published:** 2023-01-28

**Authors:** Miguel Angel Pelaez, María Florencia Torti, Aaron Ezequiel Alvarez De Lauro, Agostina Belén Marquez, Federico Giovannoni, Elsa Beatriz Damonte, Cybele Carina García

**Affiliations:** 1Laboratory of Antiviral Strategies, Biochemistry Department, School of Sciences, University of Buenos Aires, IQUIBICEN, University of Buenos Aires/Consejo Nacional de Investigaciones Científicas y Técnicas, Buenos Aires 1428, Argentina; 2Ann Romney Center for Neurologic Diseases, Brigham and Women’s Hospital, Harvard Medical School, Boston, MA 02115, USA

**Keywords:** aryl hydrocarbon receptor, Junín virus, arenavirus, Argentine hemorrhagic fever, host therapeutic target

## Abstract

Junín virus (JUNV), a member of the family *Arenaviridae*, is the etiological agent of the Argentine hemorrhagic fever, an endemic disease in the rural region of Argentina lacking a specific chemotherapy. Aryl hydrocarbon receptor (AHR) is expressed in several mammalian tissues and has been indicated as a sensor of ligands from variable sources and a modulator of the cell immune response. Interestingly, recent studies have suggested that the activation or depression of the AHR signaling pathway may play a role in the outcome of diverse human viral infections. In the present report, the effect of the pharmacological modulation of AHR on JUNV in vitro infection was analyzed. An initial microarray screening showed that the AHR pathway was overexpressed in JUNV-infected hepatic cells. Concomitantly, the infection of Vero and Huh-7 cells with the JUNV strains IV4454 and Candid#1 was significantly inhibited in a dose-dependent manner by treatment with CH223191, a specific AHR antagonist, as detected by infectivity assays, real-time RT-PCR and immunofluorescence detection of viral proteins. Furthermore, the pro-viral role of AHR in JUNV infection appears to be independent of the IFN-I pathway. Our findings support the promising perspectives of the pharmacological modulation of AHR as a potential target for the control of AHF.

## 1. Introduction

Currently, there is increasing evidence supporting that the aryl hydrocarbon receptor (AHR) is an important modulator of the host immune response and this feature contributes to the differential clinical outcomes of infections at the individual and population level [1,2]. AHR is a cytoplasmic receptor ubiquitously expressed in mammalian tissues, working as a sensor of a wide variety of endogenous and exogenous ligands with different origins such as pollutants, diet, microbiome, and host metabolism [3,4,5]. Indeed, it has been strongly suggested that AHR has a major impact on the interplay between environmental factors and viral infections [1,2]. Once AHR recognizes a ligand molecule, it is translocated to the nucleus where it heterodimerizes with the aryl hydrocarbon receptor nuclear translocator and regulates the expression of xenobiotic metabolism genes (CYP1A1, CYP1A2, CYP1B1). AHR can also interact with other transcription factors, such as NF-ĸB, to modulate a wide array of signaling pathways, including those involved in the immune response [6,7,8]. AHR can modulate the expression of immunoregulatory genes and alter the differentiation pathways of inflammatory dendritic cells and T-cells [9,10,11].

Recent studies carried out by our research group have demonstrated that activation of the AHR signaling pathway plays a pro-viral role during Zika (ZIKV), dengue (DENV), [12] and severe acute respiratory syndrome coronavirus 2 (SARS-CoV-2) infections [13]. To note, it was confirmed that AHR inhibition led to diminished viral replication in these viral models, showing that AHR represents a promising antiviral strategy.

Junín virus (JUNV) is an enveloped bisegmented RNA genome virus from the *Arenaviridae* family [14]. Each RNA segment encodes for two proteins in opposite directions. On one hand, the large segment (L RNA) encodes for the RNA-dependent RNA polymerase (L) and for the matrix protein (Z). On the other hand, the small segment (S RNA) encodes for the nucleoprotein (NP) and for the glycoprotein precursor (GPC), which is cleaved by proteases to produce three subunits (GP1, GP2, and the signal peptide SSP) that are held together to form the glycoprotein complex [14].

JUNV is the etiological agent of Argentine hemorrhagic fever (AHF) [15], an endemic disease affecting the rural worker population in the humid Pampa region of Argentina. To note, a characteristic feature of AHF is the wide range of clinical symptoms presented by infected patients. Asymptomatic or mild disease, with flu-like symptoms, occurs in 70–80% of the infections. However, severe cases involve hemorrhagic and neurological complications with a mortality rate ranging from 20 to 30% in hospitalized patients [15].

Importantly, a live attenuated vaccine named Candid#1 has proved to be highly effective in preventing severe cases of AHF [15,16]. However, the only treatment available against the disease is the administration of immune plasma from convalescent patients; a therapy that has been shown to reduce the number of fatal cases only if administered within 8 days after the onset of the first symptoms [17].

Even though in recent years multiple pieces of evidence have been gathered on the effect of AHR during infections, there is still no clear evidence on its role during arenavirus infections [2]. As the rural population is constantly exposed to AHR ligands, such as pollutants from the agricultural industry such as agrochemicals, pesticides, and fossil fuel combustion by-products [18], we evaluated in this work the impact of in vitro AHR pharmacological modulation on the cellular response to JUNV infection and the potential use of AHR as a candidate target for therapeutic intervention in AHF disease.

## 2. Materials and Methods

### 2.1. Cell Lines

The following cell lines were used: Huh-7 (JCRB Cell Bank #0403) and HepG-2 ATCC HB8065, two human hepatocyte-derived carcinoma cell lines; Vero ATCC CCL-81, a monkey-derived kidney epithelial cell line. Every cell line was cultivated in Eagle’s Minimum Essential Medium (MEM) (Gibco, ThermoFisher Scientific, Waltham, MA, USA) complemented with 5% newborn calf serum (NBCS) (Gibco, ThermoFisher Scientific, Waltham, MA, USA) and supplemented with 50 µg/mL gentamycin (Sigma-Aldrich, St. Louis, MO, USA), 1 mM sodium pyruvate (Sigma-Aldrich, St. Louis, MO, USA), 2 mM L-glutamine (Gibco, ThermoFisher Scientific, Waltham, MA, USA), and buffered with HCl/NaHCO_3_. The cell lines were cultured and maintained at 37 °C with a 5% CO_2_ atmosphere.

### 2.2. Viral Strains

The following viral strains were used: the naturally attenuated IV4454 strain obtained from a mild human case of AHF [19] and the vaccine-attenuated Candid#1 strain [20]. Vero cells were used both for virus stock generation and for virus titration through standard plaque assay. All work with infectious agents was performed in Biosafety Level 2 facilities and approved by the Office of Environmental Health and Safety at the School of Sciences, University of Buenos Aires. All personnel involved with infection procedures were vaccinated with Candid-1 provided by Instituto Nacional de Enfermedades Virales Humanas Dr. J. I. Maiztegui (Pergamino, Buenos Aires, Argentina).

### 2.3. Microarray Analysis

A total of 5 × 10^5^ HepG2 cells were seeded in 25 cm^2^ culture flasks and infected with JUNV IV4454 at a multiplicity of infection (MOI) of 1 or mock-infected. RNA was extracted and purified using RNAeasy (Qiagen, Hilden, Germany) at 24 and 48 h post-infection (pi). RNA concentrations were measured using Nanodrop. Gene expression profiles of HepG2 cells were characterized using microarray analysis (Affymetrix Human U133 Plus 2.0 GeneChips) to identify host genes transcriptionally regulated by infection. The total number of host genes significantly regulated by infection was calculated for each time point by identifying genes that showed, at least, a 1.6-fold change in expression and 95% probability of being expressed differentially (*p* = 0.05). The WebGestalt software (http://www.webgestalt.org, accessed on 3 July 2020), which employs the Wikipathways database was used for Gene Ontology and pathway overrepresentation analyses. Replicate experiments were performed on different days (*n* = 3 independent experiments per condition).

### 2.4. Small Molecule Compounds

The synthetic AHR-specific competitive antagonist CH223191, 1-Methyl-*N*-[2-methyl-4-[2-(2-methylphenyl)diazenyl]phenyl]-1*H*-pyrazole-5-carboxamide (Tocris Bioscience, Bristol, United Kingdom), was solubilized in dimethyl sulfoxide (DMSO) and used at a concentration of 1.25, 2.5, 5, 10, 20, 40, and 80 µM.

The tryptophan metabolite L-kynurenine, (2S)-2-Amino-4-(2-aminophenyl)-4-oxo-butanoic acid (Sigma-Aldrich, St. Louis, MO, USA) was solubilized in DMSO and used at a concentration of 2.5, 5, 10, 20, 40, 80, 160, and 320 µM.

All treatments were performed in the absence of light, given that the compounds present photosensitivity.

### 2.5. Cell Viability Assays

For cell viability determination Huh-7 and Vero cell monolayers were grown in 96 well plates for 24 h in MEM 5% NBCS. Cell monolayers at 90% confluency were treated with different concentrations of CH223191 (1.25 µM to 80 µM) and kynurenine (5 µM to 320 µM) for 72 h. Mock was treated with the same concentration of DMSO as the highest dose of ligand treatment and was considered as 100% of cell viability. All treatments were performed in quadruplicate. Cell viability was evaluated by the 3-(4,5-dimethylthiazol-2-yl)-2,5-diphenyl tetrazolium bromide (MTT) (Sigma-Aldrich, St. Louis, MO, USA) assay. Absorbance was measured at 570 nm in a FLUOstar OPTIMA microplate reader. Cell viability % was determined by relativizing treatment absorbance results using Mock absorbance values.

Additionally, growth capability and morphology of treated cells were studied under light microscope. Pictures were taken at 72 h with a Nikon Eclipse TS100 microscope; magnification: 100×; NA: 0.25.

### 2.6. Treatments and Viral Infections

Vero and Huh-7 cell monolayers were pretreated with CH223191 and kynurenine diluted in MEM 1.5% NBCS for 1 h or 24 h, respectively, at 37 °C. CH223191 was used at concentrations ranging between 1.25 and 20 µM, while kynurenine was used at concentrations ranging between 5 and 80 µM. Then, viral infection was performed with either the IV4454 or Candid#1 strain at an MOI of 0.5. After 1 h, the inocula were removed and the cells were exposed to both compounds for 48 h at the mentioned concentrations. At 48 h pi supernatants were collected for plaque assay and cell monolayers were processed for either indirect immunofluorescence or RT-qPCR.

### 2.7. Indirect Immunofluorescence

Huh-7 monolayers treated with CH223191 or kynurenine and infected with JUNV as mentioned above were fixed at 48 h pi with methanol for 10 min at −20 °C, washed thrice with PBS, and stored in fresh PBS at 4 °C until processing. The following antibodies were used: primary mouse monoclonal anti-NP antibodies (1:200) (NA05-AG12) [21] and secondary Alexa Fluor 488 conjugated anti-mouse goat antibodies (1:200) (A-11001) (ThermoFisher Scientific, Waltham, MA, USA). Cell nuclei were stained with 4′,6-diamidino-2-phenylindole (DAPI) (1:1000) (Sigma-Aldrich, St. Louis, MO, USA). Microscopy and photography data were obtained using an Olympus IX71 fluorescence microscope with a QImaging Exi-Aqua camera attached (pixel size: 6.45 μm; photodetector area: 1392 × 1040 pixels^2^). Quantification of fluorescence data was performed using Fiji software (version 1.53v) (National Institutes of Health, Bethesda, MD, USA). Manual quantifications of DAPI-stained nucleus and NP-positive cells were performed manually through the cell counter plugin included in the Fiji software (version 1.53v). Cells were considered positive if NP viral antigen staining could be detected. Percentage of NP-positive cells was expressed as the ratio between NP-positive cells and total cells in each field and then the average percentage was determined and presented. Fluorescence intensity of NP-positive cells was obtained by measuring mean gray value of individual cells and subtracting background fluorescence. Viral foci were delimited by considering groups of NP-positive cells surrounded by NP-negative cells. Viral foci were counted on 3 different images from 3 different randomly chosen fields and the average foci size was determined by the number of cells that formed them.

### 2.8. Viral Nucleic Acid Detection

Nucleic acid extraction was carried out from infected cell monolayers by using Highway ADN/ARN PuriPrep-VIRUS kit K1501 (Inbio Highway, Buenos Aires, Argentina), according to manufacturer’s instructions. JUNV RNA viral load and cellular transcripts quantification were carried out by RT-qPCR. Reverse transcription was performed using random primers and Moloney murine leukemia virus reverse transcriptase (M-MLV RT). For fragment amplification, the following primers were used: JUNV-np (Forward: 5′-GGC ATC CTT CAG AAC ATC-3′, Reverse: 5′-CGC ACA GTG GAT CCT AGGC-3′), ahr (Forward: 5′-ATC CTT CCA AGC GGC ATA GAG ACC-3′; Reverse: 5′-CAA AGA AGC TCT TGG CTC TCA GG-3′), cyp1a1 (Forward: 5′-TTCCGACACTCTTCCTTCGT-3′; Reverse: 5′-ATGGTTAGCCCATAGATGGG-3′), β-actin (Forward: 5′-TTA GTT GCG TTA CAC CCT TTC TTG-3′; Reverse: 5′-TCA CCT TCA CCG TTC CAG TTT-3′). PCR cycle details were as follows: initial denaturalization 2 min at 95 °C; 45 amplification cycles consisting of 30 sec at 95 °C, 1 min at 58 °C, and 1 min at 72 °C; a final cycle of 5 min at 72 °C; finally, melting curves were performed. Detection was performed using MyiQ2 equipment (Bio-Rad Laboratories, Hercules, CA, USA). Results were reported as 2^−ΔΔCt^ (Fold change) and normalized using β-actin as housekeeping gene.

### 2.9. Plaque Assay

Viral yields were determined by standard plaque assay. Cell supernatants were collected and serial dilutions were prepared to infect Vero cell monolayers for 1 h at 37 °C. After viral adsorption, the inocula were removed, MEM 5% NBCS supplemented with 0.7% methylcellulose was added and cells were incubated for 1 week at 37 °C in a 5% CO2 incubator. Afterward, cell monolayers were fixed with 10% formaldehyde, washed, and stained with crystal violet. Viral titer was calculated after plaque forming unit (PFU) quantification.

### 2.10. Statistical Analysis

Microarray statistical analysis was performed using WebGestalt software (http://www.webgestalt.org/, accessed on 3 July 2020). Data are described as mean ± S.D. One-Way ANOVA with Dunnett’s post hoc test by GraphPad Prism 8 (version 8.0.1) for Windows 11 (GraphPad Software, San Diego, CA, USA) was performed to evaluate statistical significance. *p* ≤ 0.05 was considered statistically significant.

## 3. Results and Discussion

### 3.1. The AHR Pathway Is Over-Represented during JUNV Infection

The liver is one of the main targets during JUNV infection [22]. In order to elucidate the molecular mechanisms involved in a hepatocyte infection, we conducted an Affymetrix microarray screening to determine the differentially expressed genes in liver-derived human HepG2 cells infected with JUNV IV4454 for 24 or 48 h.

We utilized the Transcriptome Analysis Console software from ThermoFisher Scientific, Waltham, MA, USA to evaluate the differentially expressed genes in the JUNV-infected cells compared to the control (Figure 1a,b). A total of 266 and 313 differentially expressed genes were detected at 24 and 48 h pi, respectively (Figure 1a,b).

To further study the impact of JUNV on the cellular landscape, we utilized the WebGestalt software (http://www.webgestalt.org, accessed on 3 July 2020), which employs the Wikipathways database as a repository to perform a Gene Ontology Analysis (Figure 1c) and to determine which signaling pathways were differentially altered compared to control (Figure 1d).

Regarding the Gene Ontology Analysis from the differentially expressed genes, it was concluded that JUNV infection impacts the expression of genes related to RNA metabolism, host kinases, and lipid metabolism (Figure 1c). It is worth noting that these biological processes and molecular functions have been reported to be targeted by JUNV during its replication cycle [23].

Furthermore, Pathway Overrepresentation Analysis revealed that JUNV infection enriches the AHR signaling pathway at 48 h pi (Figure 1d) among many other pathways (*p* < 0.05). Particularly, we detected an increased expression of the AHR target gene CYP1B1, which evidences an increased activity of the AHR pathway.

Over the last few years, several studies revealed the importance of AHR as a therapeutic target during different pathological scenarios; thus, a wide variety of small compounds to modulate its activity has been developed. We decided to further study the impact of AHR modulation during in vitro JUNV infection.

### 3.2. Pharmacological Modulation of AHR Affects Viral Replication

In order to elucidate the role that AHR plays in JUNV infections, we decided to test the effects of known AHR ligands CH223191 (antagonist) and kynurenine (agonist) on in vitro infections with two different JUNV attenuated strains: IV4454 and Candid#1. Treatments and infections were performed employing Huh-7 and Vero cells. Since this last cell line lacks the ability to express and secrete interferon type I (IFN-I), its use allows determining the importance of IFN-I expression in the potential AHR mediated host–virus interplay.

Firstly, the cytotoxicity of different concentrations of both CH223291 and kynurenine was evaluated via MTT assay (Figure 2a,b) and optic microscopy observations (Figure 2c,d).

Regarding CH223191, a decrease in the cell viability and morphological alterations associated with cytotoxic effects were detected only at concentrations of 80 µM (Figure 2a,c). On the other hand, kynurenine did not induce cytotoxic effects at any concentration tested (Figure 2b,d).

In order to investigate the effect of AHR pharmacological modulation during JUNV infection, cell cultures were treated with vehicle (DMSO), CH223191, or kynurenine and then infected with JUNV for 48 h to determine viral yield. Briefly, Vero and Huh-7 cells were vehicle-treated or treated with different concentrations of the small molecule CH223191 (2.5, 5 µM, 10 µM, and 20 µM) or kynurenine (5 µM, 10 µM, 20 µM, and 40 µM) prior and post-JUNV infection with IV4454 and Candid#1 at an MOI of 0.5. After 48 h, the supernatants were harvested and used to infect Vero cells for the PFU assay (Figure 3).

The AHR blockade significantly reduced viral particle production in a dose–response manner, even with the lowest concentration of CH223191 tested. Importantly, this result was observed not only using both JUNV-attenuated strains but also in both infected cell lines (Vero and Huh7). In fact, the CH223191 treatment of JUNV-infected Vero and Huh-7 cells diminished the number of viral plaques by 93% and 97%, respectively (Figure 3a,b). These results strongly suggest that the AHR signaling pathway is an important cellular factor during JUNV infection (Figure 3a,b). On the other hand, the kynurenine administration before and after JUNV inoculation did not significantly change the viral titer obtained in comparison to viral control (Figure 3c,d).

All in all, the results evidenced for the first time that AHR is an important cellular factor during JUNV in vitro infection, implying a pro-viral role by facilitating the viral replication cycle.

### 3.3. AHR Modulation Has an Impact on JUNV Protein Expression

To further study the effects of AHR modulation on JUNV infection, we conducted an indirect immunofluorescence assay. The JUNV NP protein is the most abundant structural and functional protein within the *Arenaviridae* family. Thus, NP was chosen as an interesting staining target given that only a few studies reported the NP expression pattern of different JUNV attenuated strains. Our first step was to determine the NP distribution of both JUNV strains in our cellular models in order to compare the permissiveness of the different cell lines and the viral dissemination of both attenuated strains in these cell cultures (Figure 4).

The NP localization was exclusively cytoplasmic and exhibited a homogeneous big punctuate accumulation pattern, similar for both strains in Vero and Huh-7 cell lines (Figure 4).

Next, we evaluated by immunofluorescence the impact of the pharmacological modulation of AHR on NP expression in JUNV-infected cell cultures.

Briefly, cells were seeded over glass coverslips, pre-treated with either vehicle (DMSO), CH223191 (10 µM) or kynurenine (40 µM), and then mock-infected or JUNV-infected for 48 h. Afterward, cells were fixed and processed through an immunofluorescence assay (Figure 5).

The CH223191 administration remarkably decreased the number of NP-positive cells in both cell cultures and for both JUNV strains (Figure 5). These observations correlate with previous results shown in Figure 3a,b. The AHR blockade reduced not only the percentage of JUNV-infected cells, but also the size of viral foci. Vero cell cultures pre-treated with CH223191 and infected with either IV4454 or Candid#1 showed a 57.14% (SD ± 7.98) and 41.17% (SD ± 9.05) reduction in foci size, respectively. In addition, Huh-7 cell cultures pre-treated with the antagonist and infected with either IV4454 or Candid#1 showed a 28.57% (SD ± 8.70) and 12.50% (SD ± 9.30) reduction in foci size, respectively. On the other hand, it was observed that the treatment with kynurenine did not alter the percentage of NP-positive cells (Figure 5), or foci size (not shown), compared to non-treated infected cells.

Furthermore, a more detailed microscopic inspection showed that viral foci were larger in infected Vero cell cultures in comparison to Huh-7-infected cell cultures. We observed that the average of JUNV-infected Vero cells per foci consisted of 35 cells, while the average of JUNV-infected Huh-7 cells per foci consisted of 6 cells. This expected observation is in line with the restricted viral environment that IFN-competent cells imposed on JUNV multiplication [24].

### 3.4. AHR Suppression Reduces JUNV Viral Levels

Finally, with the aim of evaluating if an AHR blockade impacts the JUNV RNA levels, Vero and Huh-7 cells were treated with either vehicle, CH223191 (10 µM) or kynurenine (40 µM), and then were mock or JUNV-infected for 48 h. Afterward, the cell monolayers were harvested and processed for RT-qPCR to monitor *ahr*, *cyp1a1* and *np* RNA levels (Figure 6).

It was observed that CH223191-treated and JUNV-infected cells exhibited a trend toward lower *ahr* mRNA levels compared to vehicle-treated–JUNV-infected samples (Figure 6a,b). Conversely, the administration of kynurenine displayed a trend towards an enhancement of the *ahr* mRNA levels in JUNV-infected cells compared to vehicle-treated–JUNV-infected samples (Figure 6a,b). In line with these results, treatment with CH223191 showed a trend towards a reduction in *cyp1a1* mRNA levels in Huh-7 cells, while treatment with kynurenine showed opposite effects (Figure 6c). Regarding the JUNV RNA level, it was observed that the treatment with the AHR antagonist CH223191 reduced viral RNA levels in infected cells compared to vehicle-treated–JUNV-infected samples, while kynurenine-treated and JUNV-infected cells exhibited a tendency towards an increment of the viral RNA levels (Figure 6d,e).

In this work, we showed for the first time that in vitro JUNV infection induces the activation of the AHR signaling pathway in liver-derived cell cultures. Microarray analysis data showed that the AHR signaling pathway is over-expressed in JUNV-infected cell cultures at 48 h pi.

Several studies reported that AHR activation can have a variety of effects on cell physiology, impacting proliferation and immune innate responses [6,25]. In fact, in the last decade, AHR activation has been described as having IFN modulatory activity-exerting effects on cytokine secretion [26,27]. Importantly, AHR up-regulation signaling can reduce IFN-I antiviral immune responses [28]. Regarding this, we evaluated the impact of the AHR signaling pathway modulation on non-competent and competent IFN cell cultures, such as Vero and Huh-7 cellular models, using AHR antagonist and agonist small commercial molecules during in vitro JUNV infection with two different attenuated strains.

Through different approaches, it was confirmed that AHR negative modulation via pharmacological inhibition with CH223191 had antiviral activity against JUNV. After the AHR blockade, JUNV in vitro infection was found inhibited. An important decrease in viral protein expression was observed in JUNV-infected cell cultures treated with the AHR antagonist. Furthermore, the AHR blockade diminished the extracellular infectious viral particle production of both attenuated IV4454 and Candid#1 strains of JUNV studied in this work. Moreover, a trend toward a decrease in viral RNA levels in CH223191-treated cells was observed. Interestingly, these findings were observed in both Huh-7 and Vero cell lines and showed an equivalent magnitude, suggesting that the AHR pro-viral role during JUNV infection might be independent of the IFN-I signaling pathway. These results are in line with our previous observations in other viral models [13]. More studies will be needed to elucidate which step of the JUNV replication cycle is affected by the AHR blockade.

Studies showing AHR activation by anthropogenic ligands have gained particular interest due to the growing awareness involving improper environmental exploitation and its interplay with viral infection severity [2]. To note, the habitat area covered by JUNV vector rodents comprises a large territory; however, at present, AHF only affects a restricted and confined region where mainly rural activities are undertaken [29]. Furthermore, agricultural workers have been shown to be the main population at risk to suffer severe manifestations during the AHF disease. Our present work suggests that rodent/human exposure to AHR agonists might have an impact on the outcome of JUNV infection.

Although intensive efforts have been dedicated in the last decades to antiviral research against arenaviruses [30], no specific antiviral chemotherapy is currently available for the treatment of AHF and human diseases caused by other pathogenic members of *Arenaviridae*. Particularly, Lassa virus (LASV) is the agent of Lassa fever (LF), which represents a serious human threat in regions of West Africa with a very high rate of mortality [31]. At present, the only alternative treatment against LF is the off-label use of the guanosine analog ribavirin, which has been demonstrated to be partially effective in LF patients only if its administration is started within 6 days of symptom onset [32,33]. Furthermore, ribavirin may induce adverse side effects limiting the recommendation of its administration only to patients at high risk. Then, there is a real demand for new effective antivirals for therapy of arenavirus hemorrhagic fevers. AHR represents a new host target to be considered. Indeed, there are several ongoing clinical trials involving AHR inhibitors (BAY2416964, IK-175, and HP163) in the treatment of different cancers. Nevertheless, these trials are in the early stages and none focus on the antiviral potential of AHR pharmacological targeting. Noticeably, drugs directed to cell factors required in the virus multiplication cycle have regained interest in antiviral development given the chance to obtain a wide-spectrum inhibitor affecting a host target common to several human pathogens [34,35], a feature apparently associated with AHR.

In conclusion, the combined results of the present study highlight the relevance of AHR signaling pathway modulation as a potential therapeutic target against JUNV. Future studies will be needed to implement AHR targeting therapies to overcome important challenges, such as the delivery of the AHR ligands to the desired tissues and cells to minimize possible off-target AHR modulation effects.

## Figures and Tables

**Figure 1 viruses-15-00369-f001:**
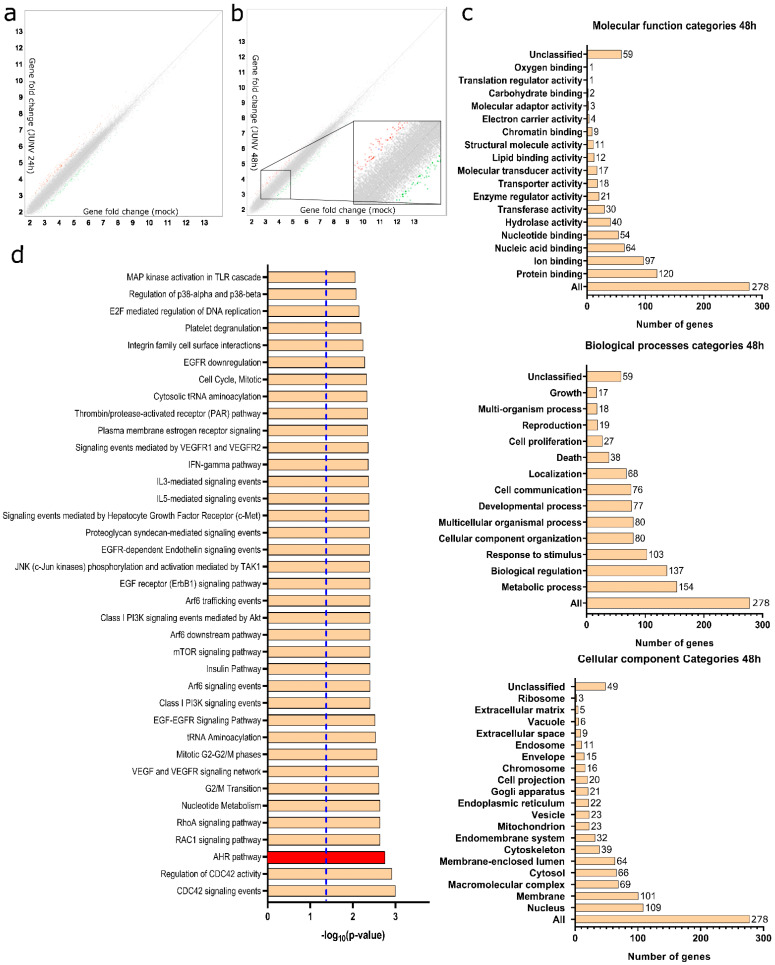
Differential gene expression assessment of JUNV-infected HepG2 cells: (**a**) Differentially expressed genes between mock-infected and JUNV-infected HepG2 cells at 24 h pi identified by Affymetrix microarray assay (*n* = 3 independent experiments per condition). Host genes showing at least a 1.6-fold change in expression and a 95% probability of being expressed differentially (*p* = 0.05) were considered for further analysis. Genes shown in red were up-regulated, genes shown in green were down-regulated, and genes in gray showed no change in expression compared to uninfected HepG2 cells. (**b**) Differentially expressed genes between mock-infected and JUNV-infected HepG2 cells at 48 h pi. (**c**) Gene Ontology Analysis of differentially expressed genes in JUNV-infected HepG2 cells at 48 h pi. (**d**) Pathway Overrepresentation Analysis of JUNV-infected HepG2 cells compared to mock-infected cells showing the main signaling pathways affected during infection. Red bar highlights the AHR pathway. Dashed blue line indicates *p* = 0.05. *p* values were determined by WebGestalt software (http://www.webgestalt.org, accessed on 3 July 2020).

**Figure 2 viruses-15-00369-f002:**
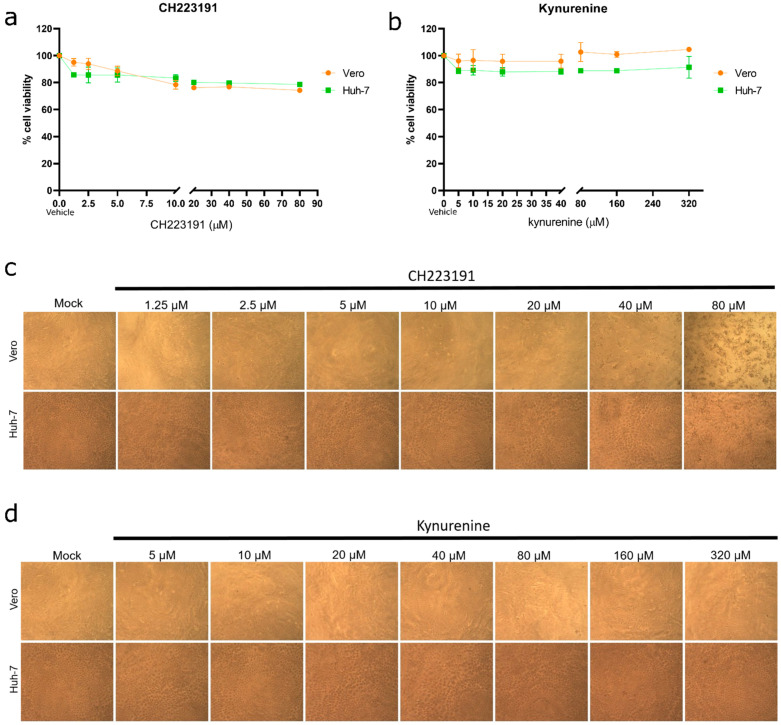
Compound cytotoxicity determination. Cell cultures were exposed to different concentrations of either CH223191 (**a**,**c**) or kynurenine (**b**,**d**) for 72 h. Afterward, viability determination of treated cells was performed by MTT assay (**a**,**b**). Mock: cells treated with vehicle. Bars represent mean ± SD. Light microscopy observations of cell cultures were performed at 72 h of exposition to both compounds (**c**,**d**). Magnification: 100×.

**Figure 3 viruses-15-00369-f003:**
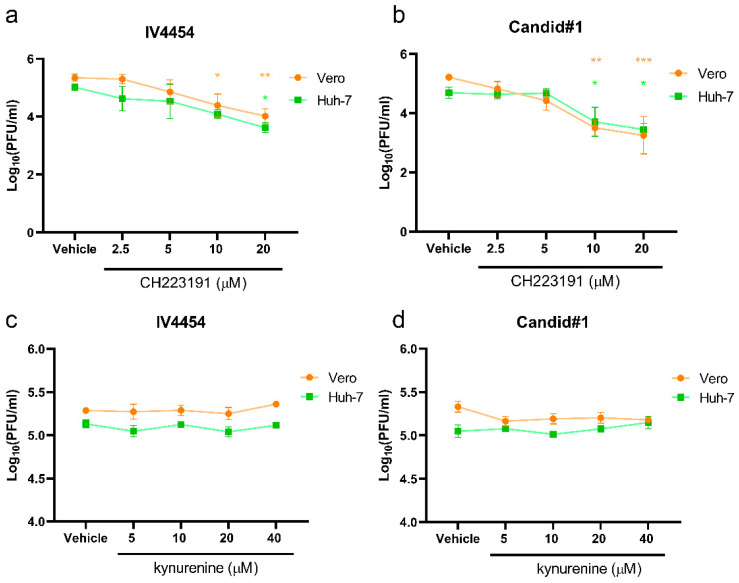
Viral yield determination through standard plaque assay of JUNV-infected cells treated with AHR ligands. Cell cultures were pre-treated with either CH223191 (**a**,**b**) or kynurenine (**c**,**d**). Subsequently, cells were either mock infected or infected with JUNV strains IV4454 (**a**,**c**) or Candid#1 (**b**,**d**). Supernatants were collected at 48 h pi. Data represent the mean ± SD. Statistical analysis was performed by comparing treatments with vehicle control. *p* < 0.05 was considered statistically significant. * *p* < 0.05; ** *p* < 0.01; *** *p* < 0.001.

**Figure 4 viruses-15-00369-f004:**
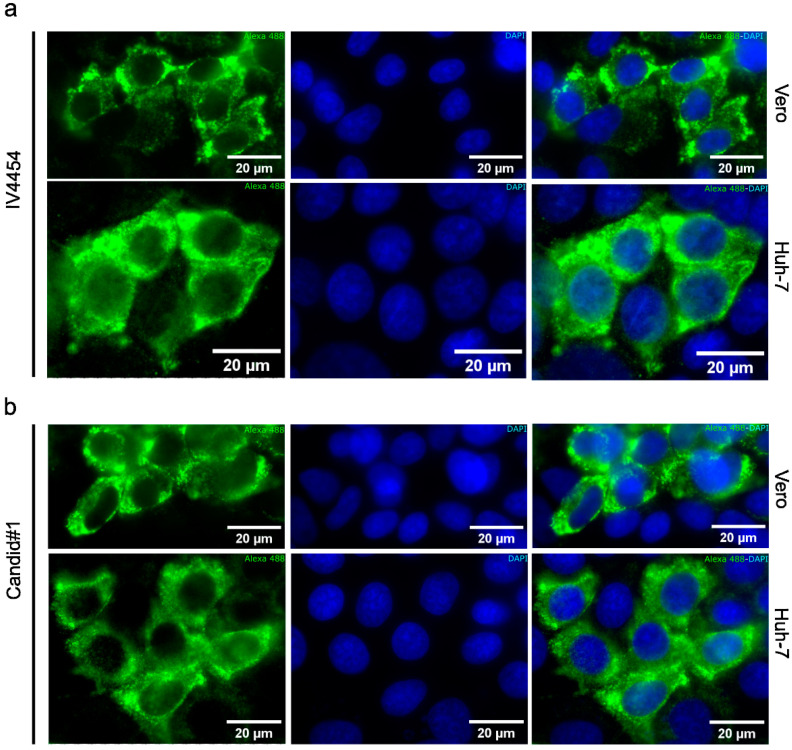
JUNV NP protein distribution on infected cells. Vero and Huh-7 cells were infected with IV4454 (**a**) and Candid#1 (**b**) (MOI = 0.5) for 48 h. Then cells were fixed and processed through immunofluorescence. NP was stained with Alexa 488 (green), and cell nuclei were stained with DAPI (blue). Scale bar: 20 µm. Magnification: 600×.

**Figure 5 viruses-15-00369-f005:**
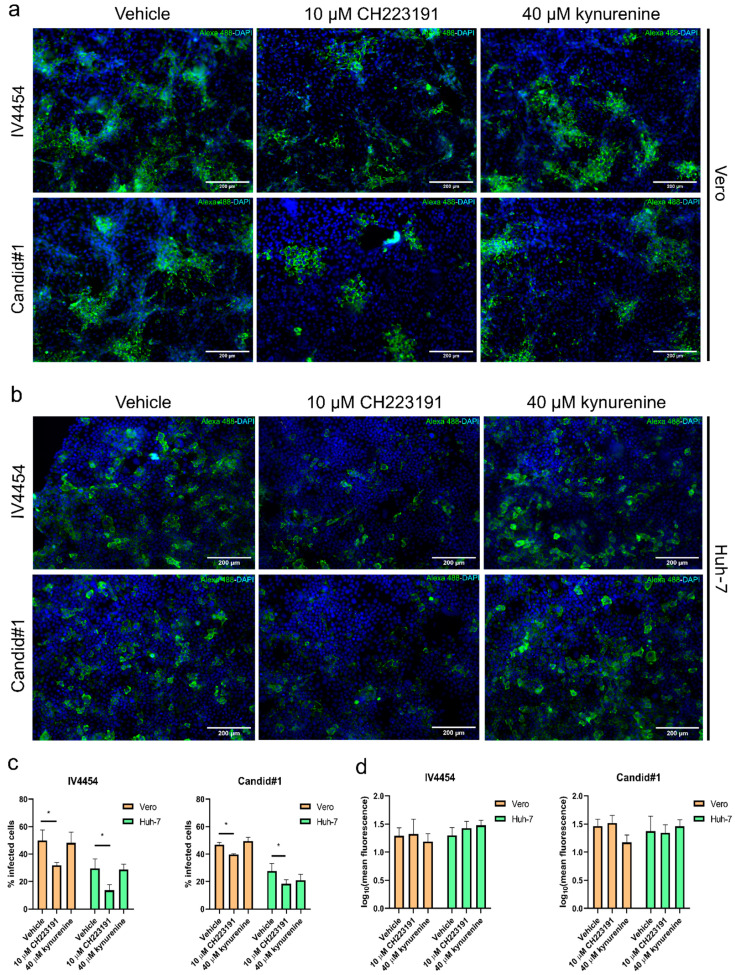
AHR impacts on JUNV protein expression. Vero (**a**) and Huh-7 (**b**) cells were infected with either IV4454 or Candid#1 (MOI = 0.5) for 48 h. Then, cells were fixed and processed through immunofluorescence. NP was stained with Alexa 488 (green), and cell nuclei were stained with DAPI (blue). Scale bar: 200 µm. Magnification: 600×. (**c**) Quantification of NP-positive cells. Results were shown as % of infected cells (% of NP-positive cells). Quantification was performed for over 200 cells per condition. (**d**) Quantification of mean fluorescence intensity of individual cells (100 cells) per treatment. Results were plotted as the mean ±SD. Statistical analysis was performed by comparing treatments with vehicle control. *****
*p* < 0.05 was considered statistically significant.

**Figure 6 viruses-15-00369-f006:**
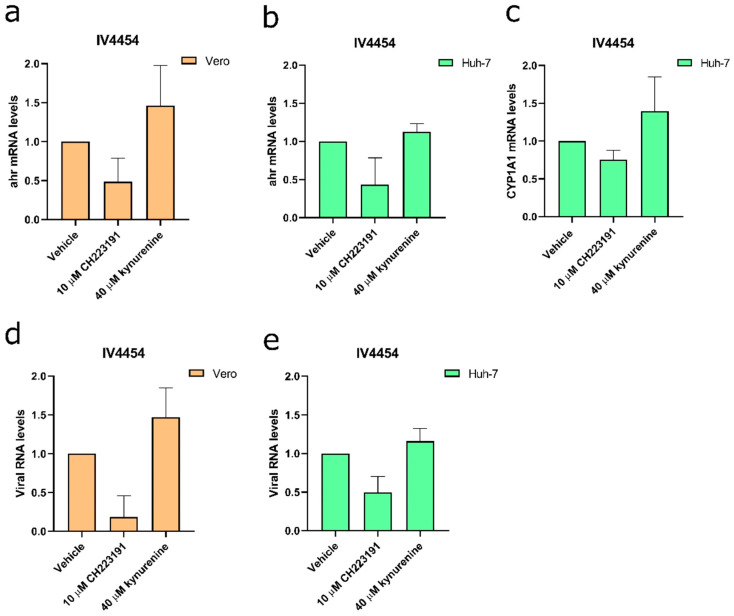
Relative RNA levels of treated and JUNV-infected Vero and Huh-7 cells. Vero cells (**a**,**d**) and Huh-7 cells (**b**,**c**,**e**) were either treated or untreated, and infected with JUNV IV4454 (MOI = 0.5) for 48 h. The results correspond to the *ahr* mRNA levels (**a**,**b**), *cyp1a1* (**c**), or *viral* RNA levels (**d**,**e**). Results were plotted as fold change relative to vehicle control. β-actin was used as a housekeeping gene.

## Data Availability

The data that support the findings of this study are available from the corresponding author upon reasonable request.

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
