# Peer review of "Modulation of the Aryl Hydrocarbon Receptor Signaling Pathway Impacts on Junín Virus Replication"

_viruses, 2023, doi:10.3390/v15020369_

Round 1

Reviewer 1 Report

The manuscript by Pelaez and coworkers reports on experiments designed to assess the effect of increasing or decreasing AHR signaling on the JUNV replication. The authors find that AHR signaling pathway supports JUNV infection and inhibiting the pathway negatively affects the output of infectious virus and reduces NP production. The findings are of general interest and add to a growing list of viral infections affected by AHR signaling. Overall, this is an interesting story but there are many issues that need to be addressed. In addition, the manuscript should be carefully edited for numerous grammatical and typographical errors that would greatly improve the reading experience of the target audience.

Major comments:

Figure 3 – Missing from this study are basic experiments to show/confirm that CH223191 and kynurenine are impacting ARH signaling in the cells. The CH compound appears to be having a biological effect as cell viability is impacted as the concentration of the drug increases. However, kynurenine treatment shows no effect on cell health even at the highest doses tested, suggesting that it may not be upregulating AHR signaling. The activity of these compounds should be verified in the cell lines used to convincingly demonstrate that the effects are indeed associated with the modulation of AHR signaling. 

Figure 5 and related text – The impact of treatments is only measured in positively stained cells vs. unstained cells (no evidence of NP staining ). However, the reduction or increase of staining intensity as a measure of the effect of treatment is not considered. For example, one treatment may reduce the intensity/volume of staining in an infected cell by 90% (with 10% staining still present) while a second treatment could reduce the staining by only 30%. The current analysis would not account for this difference.  I think both measures are important to fully understand the effect of treatments on JUNV replication in this experimental system.

Figure 6 - Why is viral RNA only looked at in Vero cells? The Huh-7 cells were used for the infectious virus and NP staining experiments, but they are noticeably missing here.

Can NP mRNA be distinguished from genomic RNA? To be consistent with NP protein expression, the authors would want to measure NP mRNA - not genomic RNA.

Toxicity to the host is always a major concern with host-directed therapies that alter important biological processes. What is the toxicity associated with impacting the AHR pathway in vivo? Has CH223191 been in mice or other species and were the plasma concentrations reached a level that reduced JUNV loads in vitro? If this information is available, it should be added to the end of the discussion.

Minor comments:

Line 81 - Authors should describe the biosafety level or risk group of the JUNV strains used

Line 87 - Was the experiment repeated on 3 different days to confirm results or were there 3 biological replicates included to assess the variability of the assessment from flask to flask? The Figure 1 legend indicates n=3, but it is not clear if the experiment was done on 3 different days or if 3 experimental replicates were included on the same day. This description should be provided in the methods section.

Lines 129-132 - Confusing - says cultured for 48 h, but the next statement says processed at 24 h.  This should be 48 h, correct?

Lines 138-139 - Which anti-NP Ab?  There are many described in the Sanchez paper.  Were they obtained through BEI Resources or somewhere else? Catalog #'s should be provided for both primary and secondary Abs as these are critical reagents to reproduce the assay.

Figure 1 - Text/numbers very difficult to read - fonts need to be maximized and the figure made larger and clearer. X and Y axes on panels a and b - what do numbers mean?

Figure 1 legend - Unclear what the metrics of panel d are conveying. Need clarification in the results text or the legend.

The (n=3 independent experiments per condition) description in line 224 is redundant and not needed - already included in line 217.

If WebGestalt software was used for the statistical analysis, this should be indicated in section 2.10 of the methods section.

Why was the GeneChip analysis done in HepG2 cells and the other work performed in Huh-7 cells? The rationale should be included in the discussion.

Figure 2 - Increase the image size to improve clarity

Figure 2 legend/line 242 – Mock: non-treated cells? Does this mean no treatment or treatment with the vehicle containing the same % of DMSO as the highest dose? Should mock be replaced with 0 µM to better reflect what was done? This should be clearly defined in the methods section.

Figure 3 - Cosmetic detail; panel b is larger than the other 3 panels

 Lines 317-320 - This text is out of place and belongs in the methods section

Lines 329-330 – For clarity, consider changing the text to “ … NP-positive cells (Figure 5) or foci size (not shown) compared to …”

Figure 6 - Are the differences in any of the RNA data statistically significant? No asterisks or indications of significant differences are shown (though it appears that the AHR message in treatment groups compared to the vehicle may be significant). Without demonstrating that the results are significant, statements need to be revised to indicate “trends”.

Lines 374-375 - Genome equivalents were not measured and therefore this statement needs to be revised to viral RNA. Moreover, there was no statistical significance indicated in the results making this statement questionable.

Lines 380-382 - Disagree with this statement. I can only see a trend with the viral RNA, not the viral yield. All in all, no significant difference in viral yield, NP staining, or viral RNA.

Author Response

Buenos Aires, January 16, 2023

Dear Editor       

Ms. Svea Quan

Thank you for your kind message with the reviewer comments of our manuscript entitled ‘Modulation of the aryl hydrocarbon receptor signaling path-way impacts on Junín virus replication’ by Miguel Ángel Peláez, María Florencia Torti, Aaron Ezequiel Alvarez De Lauro, Agostina Belén Marquez, Federico Giovannoni, Elsa Beatriz Damonte and Cybele Carina García.

We are grateful for giving us the opportunity to revise and resubmit our manuscript guided by these constructive comments raised by the reviewers.

Please find below a point-by-point response to specific reviewer´s comments that was also submitted in the respective electronic form. A new manuscript thoroughly edited including new information in the text was prepared according to both reviewer’s suggestions. All changes are highlighted along the text.

This manuscript is original and it has not been published or accepted for publication, either in whole or in part, in any form. No part of the manuscript is currently under consideration for publication elsewhere and it will not be submitted for publication elsewhere until a decision on its publication has been made in this journal.

We hope that now, as a revised version, this work will be suitable for publication in Viruses.

Looking forward to hearing from you soon,

Kind regards,

Prof. Dr. Elsa Beatriz Damonte

Reviewer´s comments to the Authors

Comments to the Author

Reviewer 1

(x) I would not like to sign my review report

( ) I would like to sign my review report

English language and style

( ) English very difficult to understand/incomprehensible

( ) Extensive editing of English language and style required

(x) Moderate English changes required

( ) English language and style are fine/minor spell check required

( ) I don't feel qualified to judge about the English language and style

        Yes        Can be improved        Must be improved        Not applicable

Does the introduction provide sufficient background and include all

relevant references?        (x)        ( )        ( )        ( )

Are all the cited references relevant to the research?        (x)        ( )        ( )        ( )

Is the research design appropriate?        ( )        ( )        (x)        ( )

Are the methods adequately described?        ( )        ( )        (x)        ( )

Are the results clearly presented?        ( )        ( )        (x)        ( )

Are the conclusions supported by the results?        ( )        ( )        (x)        ( )

Comments and Suggestions for Authors

The manuscript by Pelaez and coworkers reports on experiments designed to

assess the effect of increasing or decreasing AHR signaling on the JUNV

replication. The authors find that AHR signaling pathway supports JUNV

infection and inhibiting the pathway negatively affects the output of

infectious virus and reduces NP production.

The findings are of general interest and add to a growing list of viral infections affected by AHR

signaling. Overall, this is an interesting story but there are many issues

that need to be addressed. In addition, the manuscript should be carefully

edited for numerous grammatical and typographical errors that would

greatly improve the reading experience of the target audience.

Major comments:

Figure 3 – Missing from this study are basic experiments to show/confirm

that CH223191 and kynurenine are impacting ARH signaling in the cells. The

CH compound appears to be having a biological effect as cell viability is

impacted as the concentration of the drug increases. However, kynurenine

treatment shows no effect on cell health even at the highest doses tested,

suggesting that it may not be upregulating AHR signaling. The activity of

these compounds should be verified in the cell lines used to convincingly

demonstrate that the effects are indeed associated with the modulation of

AHR signaling.

We thank the reviewer for pointing out this observation. For modulating AHR pathway, we used 2 well characterized commercial compounds. Following the reviewer suggestion, we have included a new figure (Figure 6b,c) that demonstrate the impact of these drugs in our cellular models.

Figure 5 and related text – The impact of treatments is only measured in

positively stained cells vs. unstained cells (no evidence of NP staining

). However, the reduction or increase of staining intensity as a measure

of the effect of treatment is not considered. For example, one treatment

may reduce the intensity/volume of staining in an infected cell by 90%

(with 10% staining still present) while a second treatment could reduce

the staining by only 30%. The current analysis would not account for this

difference.  I think both measures are important to fully understand the

effect of treatments on JUNV replication in this experimental system.

We thank the reviewer for this suggestion. In this new version, we have added the fluorescence intensity analysis to fully understand the effect of the treatments on JUNV replication (Figure 5d).

Figure 6 - Why is viral RNA only looked at in Vero cells? The Huh-7 cells

were used for the infectious virus and NP staining experiments, but they

are noticeably missing here.

 We thank the reviewer for this observation. In this new version, we have added experiments performed with Huh-7 cell cultures (Figure 6e).

Can NP mRNA be distinguished from genomic RNA? To be consistent with NP

protein expression, the authors would want to measure NP mRNA - not

genomic RNA.

We thank the reviewer for this observation. Using this methodology, it is not possible to distinguish NP mRNA from genomic viral RNA.  In this new version we have corrected the figure (Figure 6d,e).

Toxicity to the host is always a major concern with host-directed

therapies that alter important biological processes. What is the toxicity

associated with impacting the AHR pathway in vivo? Has CH223191 been in

mice or other species and were the plasma concentrations reached a level

that reduced JUNV loads in vitro? If this information is available, it

should be added to the end of the discussion.

We thank the reviewer for this comment. Yes, in our previous study we have evaluated the therapeutic potential of AHR inhibition using CH223191 in pregnant SJL mice infected with ZIKV. This publication is cited on this manuscript. Indeed, we are planning to evaluate in the future the effect of CH223191 on in vivo JUNV experimental model.

Minor comments:

Line 81 - Authors should describe the biosafety level or risk group of the

JUNV strains used

We thank the reviewer for this comment. This information was included.

Lines 92-96.

Line 87 - Was the experiment repeated on 3 different days to confirm

results or were there 3 biological replicates included to assess the

variability of the assessment from flask to flask? The Figure 1 legend

indicates n=3, but it is not clear if the experiment was done on 3

different days or if 3 experimental replicates were included on the same

day. This description should be provided in the methods section.

We thank the reviewer for this comment. This description was clarified. Lines 109-110.

Lines 129-132 - Confusing - says cultured for 48 h, but the next statement

says processed at 24 h.  This should be 48 h, correct?

We thank the reviewer for this comment. This information was corrected. Line 144.

Lines 138-139 - Which anti-NP Ab?  There are many described in the Sanchez

paper.  Were they obtained through BEI Resources or somewhere else?

Catalog #'s should be provided for both primary and secondary Abs as these

are critical reagents to reproduce the assay.

We thank the reviewer for this comment. This information was detailed. Lines 151-152.

Figure 1 - Text/numbers very difficult to read - fonts need to be

maximized and the figure made larger and clearer. X and Y axes on panels a

and b - what do numbers mean?

We thank the reviewer for pointing this out. We have corrected the figure 1 according to his/her suggestion.

Figure 1 legend - Unclear what the metrics of panel d are conveying. Need

clarification in the results text or the legend.

We thank the reviewer for this comment. This was clarified. Lines 221-224/ 239-240.

The (n=3 independent experiments per condition) description in line 224 is

redundant and not needed - already included in line 217. We thank the reviewer for this comment. This was clarified.

If WebGestalt software was used for the statistical analysis, this should

be indicated in section 2.10 of the methods section. We thank the reviewer for this comment. This was added. Line 197.

Why was the GeneChip analysis done in HepG2 cells and the other work

performed in Huh-7 cells? The rationale should be included in the

discussion. 

We thank the reviewer for this comment. The rationale is mentioned at the beginning of Figure 1. The liver is one of the main targets during JUNV infection. In order to elucidate the molecular mechanisms involved in hepatocyte infection we first used HepG2 cells to perform the microarray approach, but unfortunately we lost this cellular model during the pandemic, thus we then switched to Huh-7.

As it can be seen in the figure below (data not shown in the manuscript), in our laboratory JUNV replication kinetic are identical in both hepatocyte cell lines: HepG2 and Huh-7.

Figure 2 - Increase the image size to improve clarity. This figure was improved.

Figure 2 legend/line 242 – Mock: non-treated cells? Does this mean no

treatment or treatment with the vehicle containing the same % of DMSO as

the highest dose? Yes, we are referring to the treatment with the vehicle containing the same % of DMSO as the highest dose. This was clarified in the text. Lines 127-128.

Should mock be replaced with 0 µM to better reflect what

was done? This should be clearly defined in the methods section. Now it is clarified in the text.

Figure 3 - Cosmetic detail; panel b is larger than the other 3 panels

We thank the reviewer for this comment. We have corrected this figure.

Lines 329-330 – For clarity, consider changing the text to “ … NP-positive

cells (Figure 5) or foci size (not shown) compared to …”

We thank the reviewer for this comment. We have modified the text according to the reviewer’s suggestion. Lines 342-343.

Figure 6 - Are the differences in any of the RNA data statistically

significant? No asterisks or indications of significant differences are

shown (though it appears that the AHR message in treatment groups compared

to the vehicle may be significant). Without demonstrating that the results

are significant, statements need to be revised to indicate “trends”

We thank the reviewer for this observation. We have modified the text according to the reviewer’s suggestion. Lines 358-364.

Lines 374-375 - Genome equivalents were not measured and therefore this

statement needs to be revised to viral RNA. Moreover, there was no

statistical significance indicated in the results making this statement

questionable.

We thank the reviewer for this observation. We have modified the text according to the reviewer’s suggestion. Lines 394-399.

Lines 380-382 - Disagree with this statement. I can only see a trend with

the viral RNA, not the viral yield. All in all, no significant difference

in viral yield, NP staining, or viral RNA. We thank the reviewer for pointing this out. This sentence was removed.

Reviewer 2 Report

In this study, the authors reported that inhibition of AHR signaling pathway reduce the JUNV replication. They investigated whether the AHR signaling pathway affected JUNV replication using its antagonist and agonist. The results showed that inhibition of the AHR decreased the growth kinetics of JUNV, N-protein expression, and viral RNA. The results were straightforward and demonstrated that the AHR pathway affects JUNV replication in vitro, but no further mechanisms were investigated.

Major comments

1) There is little description of the AHR signaling pathway in the introduction and discussion, and how it acts on the immune system downstream of the AHR pathway needs to be described in more detail.

2) In this study, JUNV replication was decreased even in Vero cells with an inactive IFN system, due to the influence of AHR antagonist. The discussion should be more in depth, citing references.

3) Lines 385-387. The natural host habitat may indeed be vast. However, it is possible that the habitat of rodents carrying JUNV is mainly concentrated in rural areas, and the rodents living in other areas may not carry JUNV.

4) It has not been investigated which step of JUNV replication is affected by the AHR signal. Nor has it been discussed.

Minor comments

1) Lines 332-335. Does this experiment not add antagonist /agonist?

2) Fig. 1. Please increase the resolution as the text is unreadable.

3) It should be mentioned whether AHR inhibitors are already in clinical use somewhere as a treatment for another disease, if so.

Author Response

Reviewer 2

( ) I would not like to sign my review report

(x) I would like to sign my review report

English language and style

( ) English very difficult to understand/incomprehensible

( ) Extensive editing of English language and style required

( ) Moderate English changes required

( ) English language and style are fine/minor spell check required

(x) I don't feel qualified to judge about the English language and style

        Yes        Can be improved        Must be improved        Not applicable

Does the introduction provide sufficient background and include all

relevant references?        ( )        ( )        (x)        ( )

Are all the cited references relevant to the research?        ( )        (x)        ( )        ( )

Is the research design appropriate?        (x)        ( )        ( )        ( )

Are the methods adequately described?        (x)        ( )        ( )        ( )

Are the results clearly presented?        (x)        ( )        ( )        ( )

Are the conclusions supported by the results?        ( )        (x)        ( )        ( )

Comments and Suggestions for Authors

In this study, the authors reported that inhibition of AHR signaling

pathway reduce the JUNV replication. They investigated whether the AHR

signaling pathway affected JUNV replication using its antagonist and

agonist. The results showed that inhibition of the AHR decreased the

growth kinetics of JUNV, N-protein expression, and viral RNA. The results

were straightforward and demonstrated that the AHR pathway affects JUNV

replication in vitro, but no further mechanisms were investigated.

Major comments

1) There is little description of the AHR signaling pathway in the

introduction and discussion, and how it acts on the immune system

downstream of the AHR pathway needs to be described in more detail.

We thank the reviewer for this comment. We have added more detailed information in the introduction about the interplay between AHR and the immune system. Lines 37-43.

2) In this study, JUNV replication was decreased even in Vero cells with

an inactive IFN system, due to the influence of AHR antagonist. The

discussion should be more in depth, citing references.

We thank the reviewer for this comment. We have added more discussion about the impact on AHR modulation in JUNV infected Vero cells. Lines 402.

3) Lines 385-387. The natural host habitat may indeed be vast. However, it

is possible that the habitat of rodents carrying JUNV is mainly

concentrated in rural areas, and the rodents living in other areas may not

carry JUNV. We thank the reviewer for this observation. Actually, it is a very enigmatic and unsolved question. It has been recently reported the results of an ecoepidemiological surveillance of infection of Calomys musculinus (JUNV reservoir), and there were detected antibodies in these animals living in Cordoba province where there are no confirmed cases of AHF since the 1990s. This study is now mentioned in the new version. Reference #29.

Calderón, G. E., Provensal, M. C., Martin, M. L., Brito Hoyos, D. M., García, J. B., Gonzalez-Ittig, R. E., & Levis, S. (2022). Cocirculación de virus Junin y otros mammarenavirus en área geográfica sin casos confirmados de Fiebre Hemorrágica Argentina [Co-circulation of Junín virus and other mammarenaviruses in a geographical area without confirmed cases of Argentine Hemorrhagic Fever]. Medicina, 82(3), 344–350.

4) It has not been investigated which step of JUNV replication is affected

by the AHR signal. Nor has it been discussed.

We thank the reviewer for this comment. Indeed, we are planning to evaluate in the future the effect of CH223191 on different steps of the JUNV replication cycle. This was mentioned in the discussion. Line 403.

Minor comments

1) Lines 332-335. Does this experiment not add antagonist /agonist? Yes, compounds were not added.

2) Fig. 1. Please increase the resolution as the text is unreadable. We thank the reviewer for this comment. The resolution was improved.

3) It should be mentioned whether AHR inhibitors are already in clinical

use somewhere as a treatment for another disease, if so. We thank the reviewer for this comment. This information was added in the discussion. Line 424-427.

Round 2

Reviewer 1 Report

The authors have addressed the major points raised.  I have no further comments.